# A Numerical Analysis of the Effects of Equivalence Ratio Measurement Accuracy on the Engine Efficiency and Emissions at Varied Compression Ratios

**Ruomiao Yang [1], Xiaoxia Sun [2], Zhentao Liu [1,\*], Yu Zhang [3] and Jiahong Fu [3]**

[1]    Power Machinery and Vehicular Engineering Institute, College of Energy Engineering, Zhejiang University, Hangzhou 310027, China; 12027020@zju.edu.cn

[2]    Beijing Power Machinery Research Institute, Beijing 100074, China; XiaoxiaSs201@gmail.com

[3]    Mechanical Engineering Department, Zhejiang University City College, Hangzhou 310015, China; zhangyu_gcxy@zucc.edu.cn (Y.Z.); fujh@zucc.edu.cn (J.F.)

\*    Correspondence: liuzt@zju.edu.cn

**Abstract:** Increasingly stringent regulations to reduce vehicle emissions have made it important to study emission mitigation strategies. Highly accurate control of the air-fuel ratio is an effective way to reduce emissions. However, a less accurate sensor can lead to reduced engine stability and greater variability in engine efficiency and emissions. Additionally, internal combustion engines (ICE) are moving toward higher compression ratios to achieve higher thermal efficiency and alleviate the energy crisis. The objective of this investigation was to analyze the significance of the accuracy of air-fuel ratio measurements at different compression ratios. In this study, a calibrated 1D CFD model was used to analyze the performance and emissions at different compression ratios. The results showed that carbon monoxide (CO) and nitrogen oxides (NOx) were sensitive to the equivalence ratio regardless of the compression ratio. With a slight change in the equivalence ratio, a high compression ratio had little effect on the change in engine performance and emissions. Moreover, with the same air-fuel ratio, an excessively high compression ratio (CR = 12) might result in knocking phenomenon, which increases the fluctuation of the engine output parameters and reduces engine stability. Overall, for precise control of combustion and thermal efficiency improvement, it is recommended that the measurement accuracy of the equivalence ratio is higher than 1% and the recommended value of the compression ratio are roughly 11.

**Keywords:** equivalence ratio; measurement accuracy; spark ignition engine; 1D CFD simulation; compression ratio



## 1. Introduction

Recent reports indicated that there are more than 260 million vehicles in China, most of which are powered by gasoline engines [1]. If the engine efficiency could be improved by 1%, with a mileage of 100,000 km per vehicle, it would reduce the fuel consumption by more than 10 billion liters and the mass of pollutant emissions by hundreds of millions of tons. In order to improve the performance of engines, a large number of people have studied combustion models [2–4], heat transfer [5–7], cooling systems [8–10], thermal fatigue [11], and dynamic performance [12] related to internal combustion engines, which play an important role in the national economy. However, all operating parameters of internal combustion engines (ICEs) are monitored by sensors, and the measurement accuracy of the sensors is very important for the stable operation of the engine, engine efficiency, and emissions [13–15]. In terms of emissions, the precise control of a sensor contributes significantly to the reduction of pollutant emission concentrations. Emissions have caused great harm to the environment, such as the Photochemical smog disaster in San Francisco and the haze in Beijing. Moreover, the affinity of CO and heme protein, which is used

as an oxygen carrier in the blood, is more than oxygen by approximately two hundred times. This results in the reduction of the oxygen-transporting capacity of the blood and brain hypoxia [16]. Nitrogen oxides (NOx) emissions are one of the main sources of acid rain, which is attributed to the nitric acid formation caused by the reaction of NOx in the atmosphere. Unburned hydrocarbons (UHC) are a mixture that pollutes the environment and may lead to an increased risk of cancer. Therefore, it is very necessary to improve the sensor sensitivity for better environment and engine efficiency. The growing concerns on more demanding emissions regulations and the energy crisis are arousing widespread concern for the investigation about energy saving and emission reduction [17]. In addition, downsizing internal combustion engines is now considered to be one of the most promising ways to improve fuel economy and meet emission regulations [18]. Increasing the compression ratio is one of the most effective means of improving the thermal efficiency and torque output of miniaturized engines [19].

In the last 7 years, the domestic transport industry has been dominated by internal combustion engines, more than 45 million of which were produced and sold. The production and sales of gasoline engines, taking a leading position in the market for a long time, were more than diesel engines by approximately five times [20]. Facing the increasingly severe environmental situation in China, stricter emission regulations will be implemented in the foreseeable future, which has accelerated the pace of research to reduce vehicle exhaust emissions [21]. A three-way catalyst attached to the end of exhaust pipes was considered to be an effective method for converting harmful pollutants into clean gases [22]. However, the conversion efficiency of TWC was mainly influenced by the equivalence ratio. Related studies have shown that most of carbon monoxide (CO), unburned hydrocarbons (UHC), and nitrogen oxides (NOx) can be converted into harmless exhaust emissions when the equivalence ratio is kept in a small variation range of stoichiometry (14.7) [16]. Therefore, many researchers have worked to find an efficient way to reduce the fluctuation of fuel-air ratio in order to reduce the environmental pollution.

The effectiveness of precise control of the equivalence ratio was mainly influenced by the accuracy of an air-fuel ratio measurement and the precision of the intelligent management system. A number of researchers have investigated various intelligent control strategies and algorithms. Wu et al. [23] proposed an observer-based model controller that can precisely control the air-fuel ratio system under different operating conditions. The results indicated that the efficiency of the control system was acceptable in terms of robustness and fast response. Khajorntraidet et al. [24] proposed a self-tuning control strategy by adjusting the amount of fuel injection in the fuel-rich zone. The control method was used to manage the equivalence ratio and showed excellent engine performance under oxygen-lean conditions. Trimboli et al. [25] used a modified control structure based on a delay-free model predictive controller, which can counteract some of the adverse effects in the system caused by time-varying delays.

The studies related to air-fuel ratio control have focused more on advanced intelligent algorithms or control system strategies, while the literature to date is scarce about equivalence ratio measurement accuracy [13]. Moreover, investigations related to the analysis of engine output response with different equivalence ratio sensor measurement accuracies are limited. However, the precision of the air-fuel ratio was primarily determined by the product of the accuracy of the advanced control strategy and the sensor measurement data. Measurement inaccuracies will lead to large variations in the air-fuel ratio, which has a significant impact on engine stability, performance, and emissions [26]. In addition, high compression ratios are a hot issue in view of the quest for better performance. Therefore, it is of great importance to investigate engine efficiency and emissions for various equivalence ratio measurement accuracies at different compression ratios. This paper demonstrated that the effect of measurement accuracy on engine efficiency and emissions was statically evaluated at varying compression ratios.

A calibrated one-dimensional CFD model was used to analyze the significance of the measurement accuracy within the range of compression ratios between 8 and 12. The

detailed information of calibration can be found in [27]. The goal of the investigation was to study the effect of air-fuel ratio measurement accuracy on engine efficiency and emissions at varied compression ratios. Furthermore, the fluctuations of engine performance and emissions indicators were also quantified based on Gt-power.

## 2. Numerical Simulation Methodology

The model utilized in this study is a single-cylinder spark ignition engine, which uses gasoline as a fuel. The engine speed is set at 2000 rpm, and the equivalence ratio measurement accuracy is controlled within 0.5%, 1%, and 2%. A calibrated 1D CFD model based on the GT-power shown in Figure 1, which consists of intake and exhaust ports, intake and exhaust runners, injectors, an engine, and a cylinder, can be used to predict the engine efficiency and emissions. The intake and exhaust ports describe end environment boundary conditions of pressure = 1 bar, temperature = 300 K, and fluid composition of air. For the intake runner and exhaust runner, the discretization length is 0.4*(cylinder bore diameter) on the intake side and 0.55*(cylinder bore diameter) on the exhaust side. The difference between the two values is due to the difference in the speed of sound as a result of the temperature difference. For injection models, a sequential injector with an imposed A/F ratio (PFI) is used to inject the fuel mixture into the cylinder. The injector defines injector delivery rate, fuel/air ratio, injection timing angle, number of holes per nozzle, and so on. The mechanical friction in the engine is calculated based on the "EngFrictionCF" model, which is an empirically derived model that states that the total engine friction is a function of peak cylinder pressure, mean piston speed, and mean piston speed squared. The object "EngCylTWall" is used to impose the wall temperature of head, piston, and cylinder walls. The in-cylinder heat transfer will be calculated using flow detail provided by the flow model "EngCylFlow", and an effective velocity is calculated and used in the calculation of a heat transfer coefficient using the Colburn analogy.

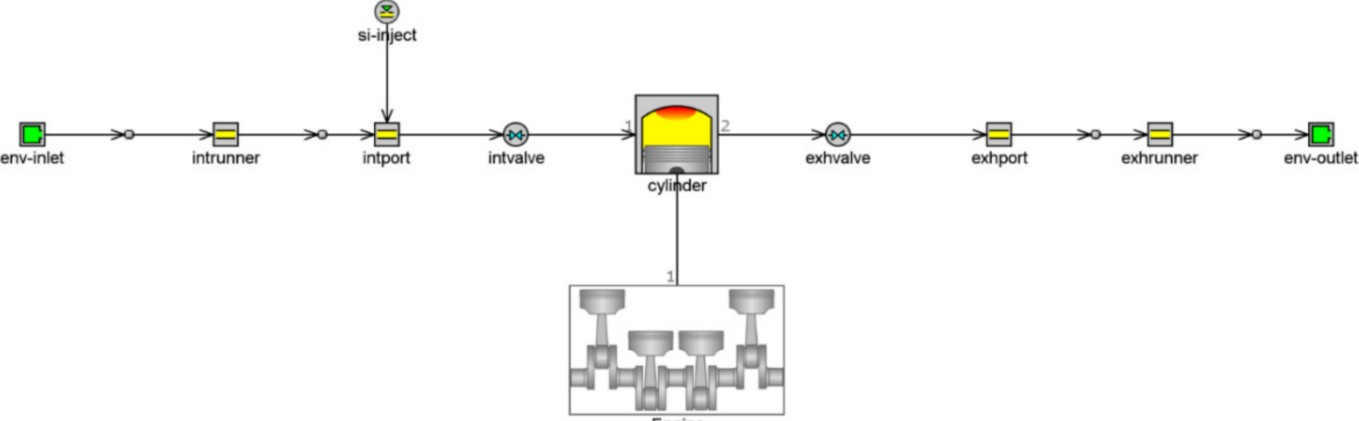

**Figure 1.** 1D CFD model of single spark ignition engine.

The basic engine parameters are shown in Table 1, and more detailed information about engine specifications can be found in [27]. The compression ratios were selected from 8 to 12 with an increment of one at the same engine speed. More simulated operating conditions are shown in Table 2. In addition, all simulations were performed at the optimal spark timing, which represents MBT timing, where MBT timing is the timing for maximum brake mean effective pressure or brake torque [28]. Figure 2 shows how MBT timing was selected—the figure illustrated the indicated mean effective pressure (IMEP) as functions of spark timing for different compression ratios at an engine speed of 2000 rpm. Moreover, the dashed line used in the graph was based on a quadratic function from the MATLAB curve fitting toolbox to fit the curve to the simulated data. In addition, Figure 2 indicated that the increase in compression ratio delays the MBT timing, as could be expected [16].

Additionally, the spark timing was delayed from $-13.3$ CAD to $-7.6$ CAD ATDC when the CR value was increased from 8 to 12.

**Table 1.** Engine specifications.

| Research Type | Single-Cylinder |
|---|---|
| Cycle | 4-stroke SI PFI |
| Valves per cylinder | 2 |
| Bore [mm] $\times$ Stoke [mm] | $86 \times 86.07$ |
| Intake valve opens | 9 CAD BTDC Exhaust |
| Intake valve closes | 96 CAD BTDC Compression |
| Exhaust valve opens | 125 CAD ATDC Compression |
| Exhaust valve closes | 38 CAD ATDC Exhaust |
| Connecting rod length [mm] | 175 |
| Piston cup diameter [mm] | 80 |
| Piston cup depth [mm] | 5 |
| Wrist pin to crank offset [mm] | 1 |
| TDC clearance height [mm] | 1 |

**Table 2.** Simulated operating conditions at same engine speed = 2000 rpm at optimal spark timing.

| Compression Ratio ($-$) | 8 | 9 | 10 | 11 | 12 |
|---|---|---|---|---|---|
| Intake pressure (bar) | 1.0 | 1.0 | 1.0 | 1.0 | 1.0 |
| Spark timing, ST (CAD ATDC) | $-13.3$ | $-11.5$ | $-9.9$ | $-8.7$ | $-7.6$ |
| Injector delivery rate(g/s) | 6.0 | 6.0 | 6.0 | 6.0 | 6.0 |
| Intake air temperature (K) | 300 | 300 | 300 | 300 | 300 |

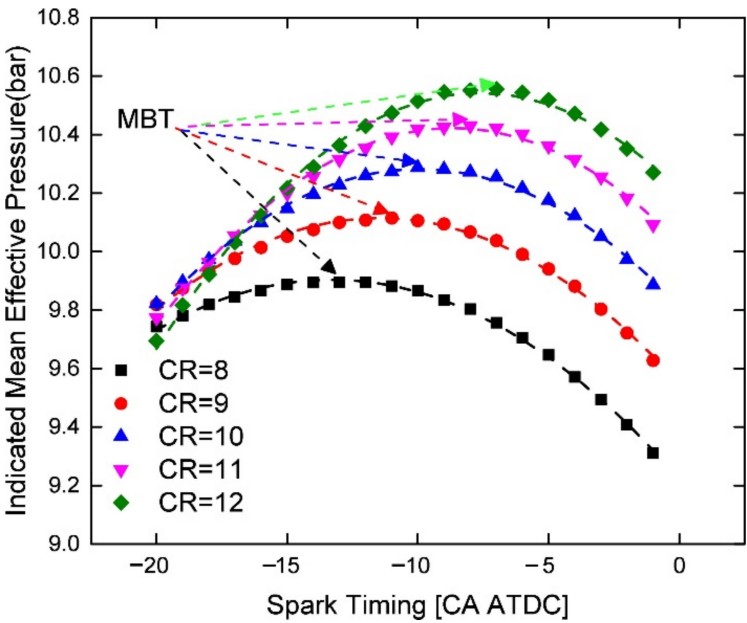

**Figure 2.** Effects of compression ratio and spark timing on indicated mean effective pressure.

The cylinder heat transfer model was Woschini, based on exponential laws, and the predictive "EngCylCombSITurb" model was utilized in the combustion process [22]. The geometry of the cylinder, spark position and timing, air motion, and fuel properties are taken into account in the predictive combustion model, which can be used to model the important physics in the combustion process in order to predict the combustion burn rate.

While nitric oxide (NO) and nitrogen dioxide ($NO_2$) are usually grouped together as NOx emissions, nitric oxide is the predominant oxide of nitrogen, accounting for more than 90% of the nitrogen oxides produced inside the engine cylinder [29]. The calculation of NO is based on the extended Zeldovich mechanism. The principal reactions governing

the formation of NO from molecular nitrogen (and its destruction) in combustion of near stoichiometric air-fuel mixtures are based on Equations (1), (2), and (3), respectively [29].

$$N_2 + O = N + NO \tag{1}$$

$$N + O_2 = O + NO \tag{2}$$

$$N + OH = H + NO \tag{3}$$

The calculations for CO are based on a kinetic model with a reaction rate mechanism that forces the number of temperature zones attribute to be "two-temp" [29]. The principal CO oxidation reaction in hydrocarbon-air flames is based on Equation (4).

$$CO + OH <=> CO_2 + H \tag{4}$$

The generation of unburned hydrocarbons (UHC) is mainly attributed to the wall quenching effect caused by the crevice volume in the combustion chamber [30]. All hydrocarbons trapped in this crevice volume at the end of combustion remain unburned, which is due to the fact that the flame cannot pass into the volume with large surface-to volume to continue burning. In summary, the 1D CFD model has been calibrated by bench test experiments, and more detailed information on the model setup and calibration can be found in [27].

## 3. Results and Discussions

### 3.1. Compression Ratios Effects on Performance

The engine efficiency and emissions were systematically evaluated at varied compression ratios (CR = 8,9,10,11,12) at a fixed engine speed = 2000 rpm with different equivalence ratio measurement accuracy around the stoichiometric ratio (14.7). It is commonly recognized that indicated specific fuel consumption (ISFC), indicated mean effective pressure (IMEP), indicated thermal efficiency (ITE), and exhaust gas temperature (EGT) are the key evaluation indicators of engine efficiency at various fluctuations in air-fuel ratio. Indicated specific fuel consumption (ISFC), which is of great interest when purchasing a car, is a significant evaluation indicator of fuel efficiency. With millions of spark ignition (SI) engines in the world, a higher ISFC means poorer fuel economy, and a small increase in ISFC can result in a significant loss of fuel economy. Nevertheless, a lower ISFC may lead to inadequate engine power output, insufficient torque, and reduction of the ability of vehicles to overcome difficult road conditions. Indicated mean effective pressure (IMEP) is a representative indicator of combustion processes and efficiency, which can be used to determine the optimum spark timing (MBT). This parameter is used to measure the indicated engine output work per unit of the cylinder working volume. A decrease in output power means a lack of braking torque, which will affect the driving operating conditions of vehicles on normal roads. Moreover, excessive horsepower represents a waste of energy, exacerbating our domestic energy crisis and contradicting the basic policy of energy conservation and pollution reduction. The indicated thermal efficiency (ITE) is inversely proportional to the indicated specific fuel consumption (ISFC), which is attributed to the product of two indicators determined by the heating value of the fuel (a constant value) [29]. With regard to the exhaust gas temperature (EGT), it is a significant parameter that affects the performance of the three-way catalyst (TWC) performance, which converts carbon monoxide (CO), nitrogen oxides (NOx), and unburned hydrocarbon (UHC) into clean gases. In addition, the lubricant properties are a key parameter in controlling the engine combustion process and engine output horsepower [31]; however, higher exhaust gas temperature will result in lower viscosity of the lubricant, which may lead to insufficient oil thickness, poor lubrication, poor sealing performance, and noisy engines. The increased viscosity of the engine oil due to the lower temperature will increase the wear of the spark ignition and fuel consumption. Therefore, the variation of these indicators must be kept within a narrow range for better engine performance. It is of great importance

to investigate the fluctuations of engine efficiency indicators for different fuel-air ratio measurement accuracies.

The effect of equivalence ratio measurement accuracy on ISFC, IMEP, and ITE is systematically evaluated and shown in Figure 3. Figure 3a–e (CR = 8,9,10,11,12) indicates that the sensitivity of ISFC and ITE to the accuracy of the equivalence ratio measurements is higher in the fuel-rich zone than in the oxygen-rich conditions. This is probably due to the fact that the increment of flame combustion speed in the fuel-lean zone is more than the reduction in the fuel-lean condition when the compression ratio is in the range of 8–12. Moreover, it is worth noting that the variation of these parameters is similar regardless of the compression ratio. In addition, Figure 3a indicates that IMEP is more sensitive to air-fuel ratio measuring accuracy in the oxygen-rich zone than in the fuel-rich condition. This is probably attributed to fuel energy loss caused by incomplete combustion in the fuel-rich zone. Compared with Figure 3b–e, the variation of IMEP, with the same measuring precision, decreases for higher compression ratios (within the range of CR = 8~12) in the fuel-rich condition. This is probably due to the fact that higher CR speeds up flame propagation and reduces the tendency of incomplete combustion. Figure 3f indicates that EGT is more sensitive to the measurement accuracy in the fuel-rich zone than in the oxygen-rich condition within the range of CR from 8 to 12, probably attributed to an incomplete combustion process. Moreover, with a compression ratio equal to 12, the fluctuation of EGT in the oxygen-rich zone is the smallest compared to the rest of the CR values. In the fuel-rich zone, the smallest fluctuation of EGT is at CR = 8. In addition, the fluctuations in the engine efficiency indicators are smaller than the changes in the measurement accuracy. Therefore, these parameters are not very sensitive to the measurement accuracy of the equivalence ratio, at least for the range of operating conditions investigated here.

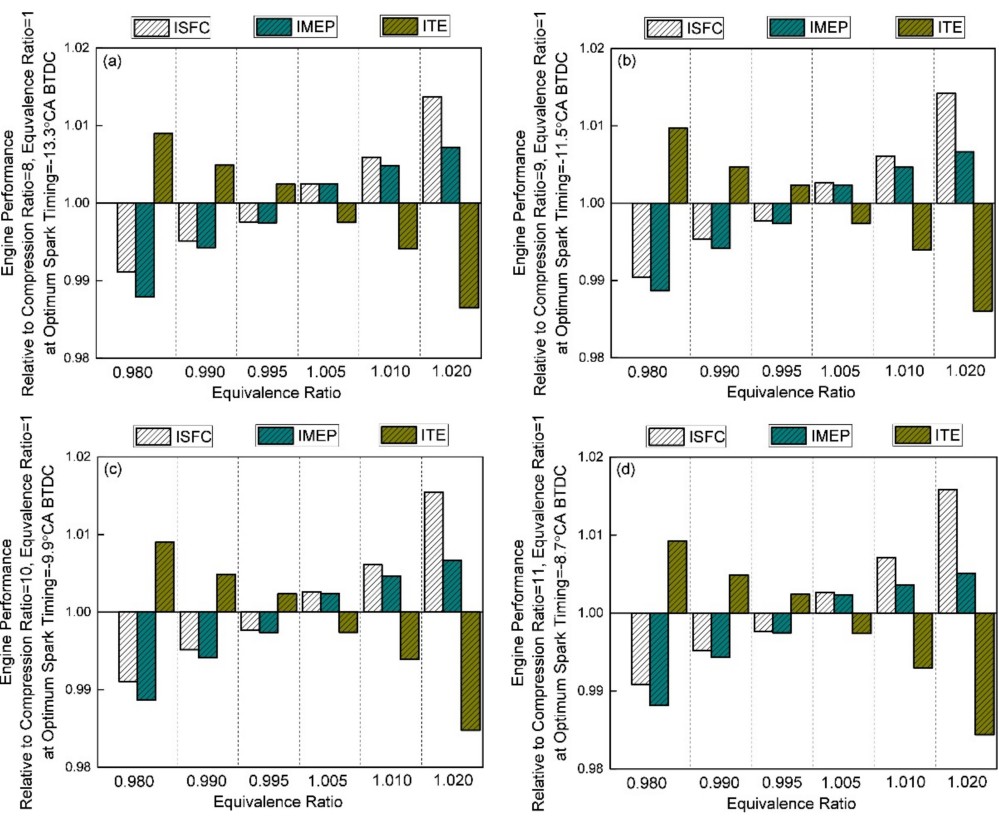

**Figure 3.** *Cont.*

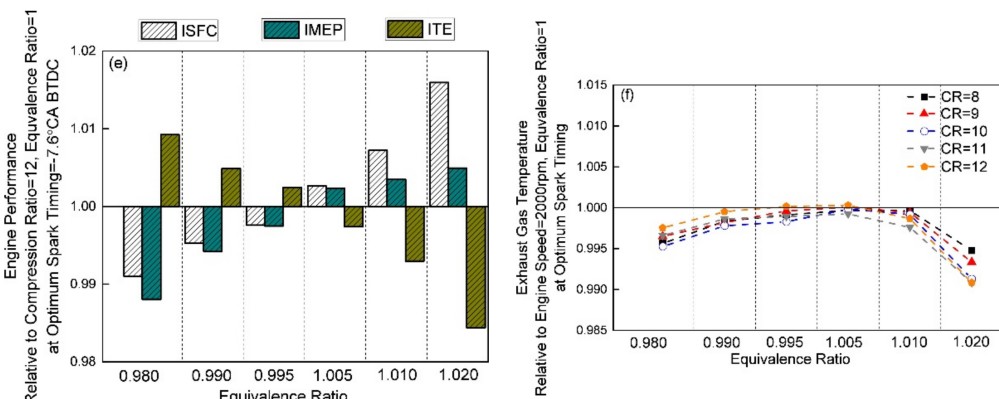

**Figure 3.** Effects of compression ratio and equivalence ratio on engine performance: (**a**) engine performance (CR = 8), (**b**) engine performance (CR = 9), (**c**) engine performance (CR=10), (**d**) engine performance (CR = 11), (**e**) engine performance (CR = 12), (**f**) exhaust gas temperature (CR = 8–12).

*3.2. Compression Ratios Effects on Emissions*

The fluctuations of emission indicators at different measurement accuracies are also quantitively evaluated in Figure 4. Carbon monoxide (CO), nitrogen oxides (NOx), and unburned hydrocarbon (UHC) are the key indicators to assess the level of engine emissions. Carbon monoxide (CO) emissions from internal combustion engines (ICEs) are primarily controlled by the air-fuel ratio [29]. The formation of NOx is mainly determined by three factors: temperature, oxygen concentration, and reaction time. The generation of hydrocarbons is mainly attributed to incomplete combustion, wall quenching, and oil film absorption. However, a trade-off relation exists: the concentration of carbon monoxide increases, and the generation of nitrogen oxides decreases from fuel-lean to fuel-rich mixtures, which is due to the fact that the combustion process can be transformed from incomplete to complete combustion within the narrow range around the stoichiometry ratio (14.7). Therefore, it is necessary to control the air-fuel ratio and improve engine stability in order to keep all emissions within an acceptable range.

Figure 4 shows the effect of equivalence ratio measurement accuracy and compression ratio on engine-out emissions. The emission indicators were systematically evaluated to find the sensitivity to the measuring precision under different compression ratios. Figure 4 indicates that CO was the most sensitive to equivalence ratio measurement accuracy, followed by NOx, while UHC was not sensitive, regardless of the compression ratio. Figure 4a shows that the variation of CO in the fuel-rich zone was larger than that in the oxygen-rich condition. This is probably attributed to the differences between the combustion process in the fuel-rich zone and fuel-lean condition. The production of CO with 2% air-fuel ratio measuring accuracy ratio was more than that with stoichiometric ratio (14.7) by approximately two times. This was probably due to the strong influence of the incomplete combustion process on CO production. Figure 4a also indicates that higher measurement accuracy in the fuel-rich zone decreased the fluctuation of NOx concentration, which was due to the reduction of incomplete combustion tendency. With 2% air-fuel ratio measurement accuracy ratio, the generation of NOx was more than the stoichiometric ratio (14.7) by about 10–15%, which means small changes in air-fuel ratio measurement accuracy had a large impact on the variation of NOx generation. With regard to unburned hydrocarbons, which may be generated due to uneven fuel mixture and an incomplete combustion process, the fluctuation of UHC was not very obvious with different measuring precisions, which means that the equivalence ratio had little effect on the production of unburned hydrocarbon (UHC). Compared with Figure 4b–e, the changes of CO decreased with higher CR value in the range of compression ratios 8–12. This is probably attributed to the fact that higher CR increased the homogeneity of the in-cylinder fuel mixture and the mixture gas temperature at the end of the compression stroke. Overall, the changes of CO and NOx were sensitive to the equivalence ratio measurement accuracy at the fixed

engine speed = 2000 rpm, while UHC was not. At the same air-fuel ratio measurement accuracy, changing the compression ratio in the range of 8–12 had little effect on the change of emission indicators.

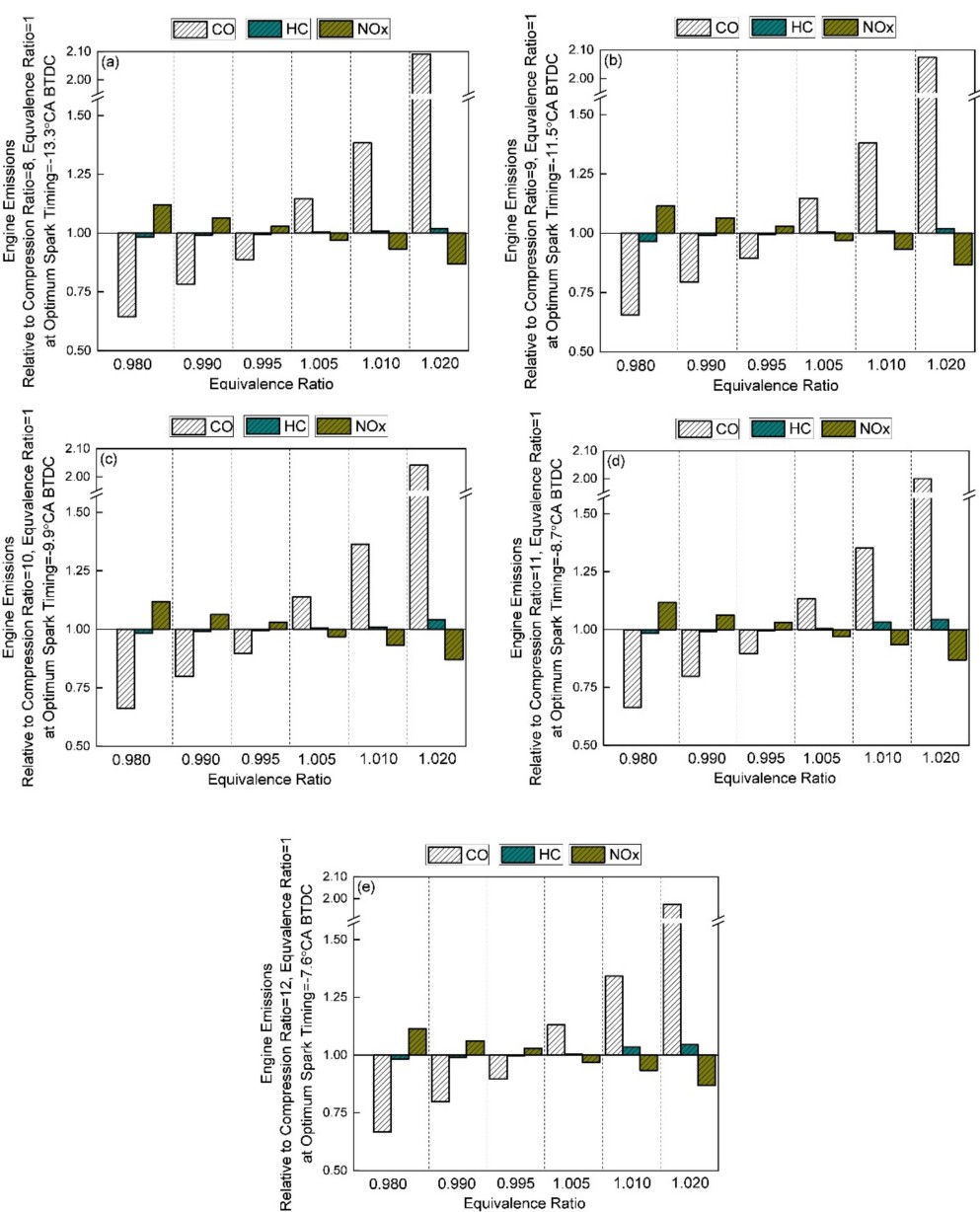

**Figure 4.** Effects of compression ratio and equivalence ratio on engine emissions: (**a**) engine emissions (CR = 8), (**b**) engine emissions (CR = 9), (**c**) engine emissions (CR = 10), (**d**) engine emissions (CR = 11), (**e**) engine emissions (CR = 12).

*3.3. Compression Ratios Effects on Combustion Phasing*

To investigate the effects of different equivalence ratio measurement accuracies on the crucial parameters describing the combustion process, Figure 5 shows the quantified assessment results of the fluctuations in ignition lag (0–2%), burn duration (0–50%), and burn duration (10–90%). Combustion phasing is a key operating indicator that greatly affects engine efficiency and emissions. Therefore, it is necessary to keep the combustion phase variation within a small range to improve engine stability and performance.

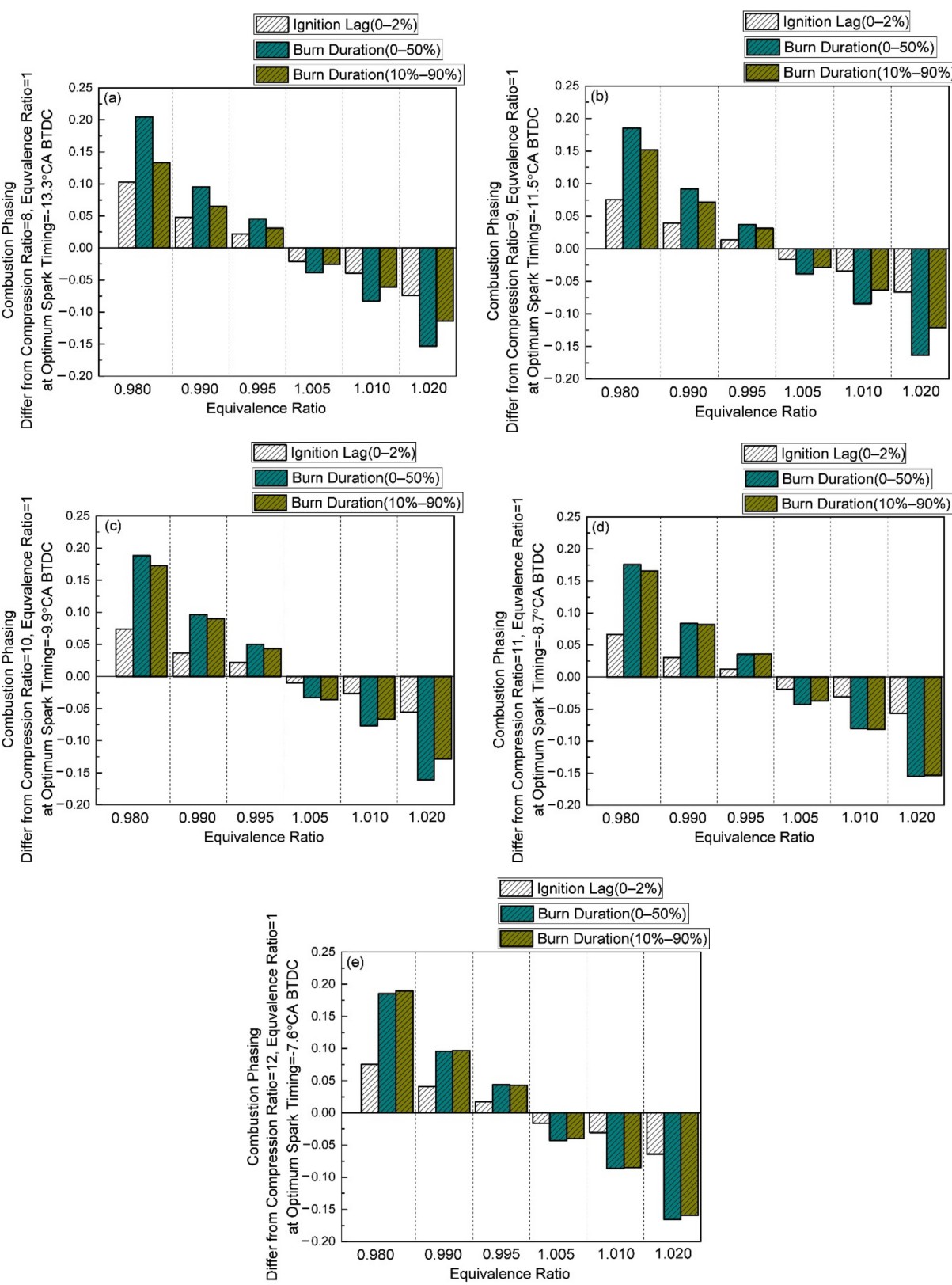

**Figure 5.** Effects of compression ratio and equivalence ratio on combustion phasing: (**a**) combustion phasing (CR = 8), (**b**) combustion phasing (CR = 9), (**c**) combustion phasing (CR = 10), (**d**) combustion phasing (CR = 11), (**e**) combustion phasing (CR = 12).

Figure 5a indicates that a higher air-fuel ratio decreases the ignition lag, burn duration (0–50%), and burn duration (10–90%), which is probably due to the increment of flame prop-

agation speed caused by a richer fuel mixture. Figure 5a also shows that the burn duration (0–50%) is the most sensitive to the air-fuel ratio, followed by burn duration (10–90%) and ignition delay. Moreover, compared with Figure 5b–e, the variation of ignition lag (0–2%), burn duration (0–50%), and burn duration (10–90%) were greater in the fuel-lean zone than in the fuel-rich zone with 2% measurement accuracy, regardless of compression ratios. This is probably attributed to the acceleration of flame front propagation speed caused by a rich fuel mixture. Moreover, in the oxygen-rich zone, as the compression ratio increased from 8 to 12, the difference between the amount of change in combustion durations 0–50% and 10–90% decreased. This was probably due to the fact that a higher compression ratio improved homogeneity of the mixture. Overall, the effect of measurement accuracy on combustion phasing was not very obvious when the CR value was in the range of 8~12.

*3.4. Measurement Accuracy Discussions*

The reduction of carbon monoxide, nitrogen oxides, and other harmful pollutants, as well as the improvement of engine output horsepower and fuel efficiency, rely mainly on the precise control of the air-fuel ratio (AFR) [32]. The actual effectiveness of controlling the air-fuel ratio is mainly determined by the control algorithm and equivalence ratio measurement accuracy. Many researchers have investigated different control methods to improve the accuracy of the algorithm, but the literature related to the measurement accuracy is limited. In addition, engines with higher compression ratios can improve energy use efficiency and alleviate environmental crisis to some extent. Therefore, it is necessary to investigate the effect of measurement accuracy on engine efficiency and emissions at varied compression ratios.

In this study, the variation of parameters, measuring engine operating conditions and performance, was quantitatively evaluated at different compression ratios with different measurement precisions. Similar findings were observed by various researchers. Wu et al. investigated the effect of the air-fuel ratio on the performance of a SI engine at a compression ratio equal to 9.5 [33]. Yontar et al. analyzed the effects of the air-fuel ratio and CNG addition for a dual sequential ignition engine with a compression ratio of 10.8 [34]. Topgül et al. studied the effects of different gasoline blends on the engine efficiency and exhaust gases by varying the compression ratio (8:1, 9:1, and 10:1) [35]. The available experiments are similar to the results investigated in this study. The results indicate that CO and NOx are sensitive to the measurement accuracy, while UHC, engine efficiency indicators, and combustion phasing are not sensitive to the measurement accuracy. The typical result at the compression ratio of 11 was chosen and shown in Figure 6, which showed the variation of sensitive parameters in relation to the air-fuel measurement accuracy. Figure 6 indicated that a 2% measurement accuracy would lead to huge errors of CO production. Moreover, the price of a 0.5% measuring precision sensor was much higher than that of a 1% measuring accuracy sensor. With 1% measurement precision, the conversion rate of a three-way catalyst is relatively high, and the variations of these sensitive indicators are within acceptable limits [16]. Therefore, a 1% measurement accuracy sensor can meet the cost and better control precision requirements. At the same time, the compression ratio can be increased as much as possible under the condition that CR has little influence on the fluctuations of other operating parameters. In addition, since the compression ratio of 12 is too high, this will have a tendency to cause engine knocking. The model used in this study cannot accurately predict the detonation phenomenon. Therefore, the measurement accuracy and compression ratio are advised to be higher than 1% and to be 11, respectively.

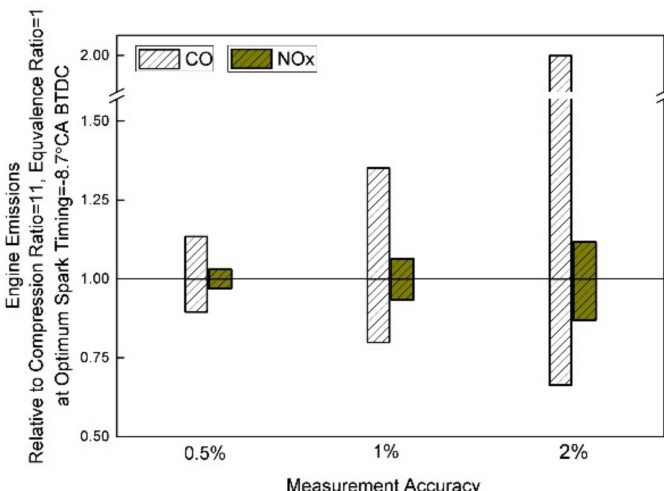

**Figure 6.** Effects of equivalence ratio measurement accuracy on emissions.

## 4. Future Work

In this paper, a single-cylinder gasoline engine model was utilized to systematically analyze the effects of equivalence ratio measurement accuracy on engine efficiency and emissions at varied compression ratios. It is generally recognized that a higher compression ratio (CR) value is a benefit for improving thermal efficiency, increasing engine power output and reducing residual exhaust gas [36,37]. However, subject to the limits of detonation phenomenon and the increment of mechanical friction loss, the value of the compression ratio cannot be too high for spark ignition engines [38]. Additionally, the CR value of high-power density engines also needs to be higher for better performance. At present, CR values can be increased by adding additives to the gasoline, such as adding 15% ethanol to increase the anti-detonating quality of the fuel mixture. In addition, methane has good anti-explosive properties. A variety of fuels will be studied in future research, such as natural gas [39,40], hydrogen blended natural gas [41,42], LPG [43,44], syngas [45,46], ethanol [47,48], and an ethanol gasoline blend [49,50]. Moreover, the sensor sensitivity will have different effects on engines fueled with various fuels, which is due to the fact that the different physicochemical properties of fuels result in different laminar flame speeds and ideal air-fuel ratios. Therefore, the conclusions in this study are only applicable to gasoline engines, and other fuels need to be studied further. In the foreseeable future, engines fueled with natural gas, LPG, ethanol, and so on, which can broaden the range of CR values, such as CR = 13 or 14, will be used to study the effects of sensor measurement accuracy on engine efficiency and emissions.

## 5. Summary and Conclusions

A calibrated one-dimensional computational fluid dynamics model was utilized to analyze the effect of equivalence ratio measurement accuracy on engine efficiency and emissions at different compression ratios. Moreover, compared to the results related to different compression ratios and measurement accuracy, the suitable sensor sensitivity and compression ratio can be chosen. The main conclusions of this investigation were:

- With small changes in equivalence ratio measurement accuracy at different compression ratios, the performance indicators changed very little. Therefore, indicated specific fuel consumption (ISFC), indicated mean effective pressure (IMEP), and indicated thermal efficiency (ITE) were not sensitive to the air-fuel ratio measurement accuracy, at least within the operating conditions investigated here.
- The air-fuel ratio measurement accuracy had a significant impact on carbon monoxide (CO) and nitrogen oxides production. In contrast, the variation of the air-fuel ratio at different compression ratios (CR = 8~11) did not have a significant effect on other engine efficiency indicators and unburned hydrocarbons (UHC).

- At the same air-fuel ratio, the variation of CO decreased with increasing CR value in the range of compression ratio 8~12. Additionally, the varied compression ratio, within the range of 8~12, had little effect on the variation of NOx.
- The compression ratio of the engine could not be increased indefinitely due to the knock presence limit, and too high CR was not beneficial for emission control. Therefore, the compression ratio should be increased, and the recommended value is 11.

Overall, the fluctuations in carbon monoxide and nitrogen oxides were sensitive to the changes in measurement accuracy, regardless of compression ratio. In contrast, the variations in other indicators describing engine efficiency, unburned hydrocarbons, and combustion phasing were less relevant to measurement accuracy, at least within the operating conditions of this study. A too high compression ratio (CR = 12) might result in the knocking phenomenon and reduce the engine's no-burst operating range, which is detrimental to the operational stability of the engine. Moreover, the compression ratio is recommended to be 11 and equivalence ratio measurement accuracy is proposed to be higher 1%.

**Author Contributions:** R.Y., conceptualization, methodology, simulation, writing—draft preparation; X.S., simulation, writing—draft preparation; Z.L., validation, writing—review and editing; Y.Z., analysis; J.F., supervision. All authors have read and agreed to the published version of the manuscript.

**Funding:** This research received no external funding.

**Institutional Review Board Statement:** Not applicable.

**Informed Consent Statement:** Not applicable.

**Acknowledgments:** Thanks for the equipment and the right to use the software provided by Power Machinery and Vehicular Engineering Institute, Zhejiang University.

**Conflicts of Interest:** The authors declare no conflict of interest.

## Abbreviations

| | |
|---|---|
| 1D | One-dimensional |
| AFR | Air-fuel Ratio |
| ATDC | After Top Dead Center |
| BTDC | Before Top Dead Center |
| CAD | Crank Angle Degree |
| CFD | Computational Fluid Dynamics |
| CO | Carbon Monoxide |
| CR | Compression Ratio |
| ICE | Internal Combustion Engine |
| IMEP | Indicated Mean Effective Pressure |
| ISFC | Indicated Specific Fuel Consumption |
| ITE | Indicated Thermal Efficiency |
| LPG | Liquefied Petroleum Gas |
| MBT | Maximum Brake Torque |
| NOx | Nitrogen Oxides |
| PFI | Port Fuel Injection |
| RPM | Revolutions per minute |
| SI | Spark Ignition |
| ST | Spark Timing |
| TDC | Top Dead Center |
| TWC | Three-way Catalyst |
| UHC | Unburned Hydrocarbon |

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
