# Peer review of "A Numerical Analysis of the Effects of Equivalence Ratio Measurement Accuracy on the Engine Efficiency and Emissions at Varied Compression Ratios"

_processes, doi:10.3390/pr9081413_

Round 1

Reviewer 1 Report

The reviewed article presents a scientific-research work focused on one of the topical scientific-technical problems that are relevant in the area of internal combustion engines, which are used as driving units in the motor-vehicles. Namely, there is presented an analysis of significance concerning the accuracy of the fuel-air ratio measurements using different compression ratios. There is applied this study a calibrated 1D CFD model in order to analyse the engine performance and exhaust gas emissions at different values of engine compression ratio. It is evident that the authors are professionally well-oriented in the given area. This article is suitable for publishing in this journal, but it requires some minor improvements. My comments and suggestions are as follows:

- The list of Abbreviations should be moved at the beginning of the article with regard to a better understanding of the used abbreviations for the readers because some of the unexplained abbreviations are occurring already in the first chapters Abstract and Introduction.

- The 2nd Chapter “Numerical simulation methodology” should be supplemented by a clearer explanation or description of the numerical simulation methodology, which was used in order to obtain the simulation results that are consequently presented in the 3rd Chapter.

- Despite the fact that the scheme of the 1D CFD model shown in Figure 1 is relatively simple, it would still deserve at least a brief description.

- My recommendation also is to transform the graphs in Figures 3, 4 and 5 into the colour form with regard to a better orientation for the readers. However, this is only an optional suggestion, which remains fully up to the authors; it is not obligatory.

I recommend publishing this article after the implementation of my comments.

Author Response

Dear reviewer: my responses to your comments and my revised manuscript are in the attachment, please see the attachment, thank you! 

Reviewer 2 Report

There are few comments on Results and Discussion part of the manuscript.

  1. Lines 198-201. How your article topic is related to discussion about the affinity of CO and heme protein etc.? Are you doing research on emission influence of human health? Please comment and conclude on that issue.
  2. Lines 201-205. The same as previous comment on NO and UHC. Please comment.
  3. Lines 236-239. The variation of NOx decreased in the oxygen-rich zone, compared the different CR values, with the same measuring accuracy at compression ratio=12. But the same decrease magnitude was noticed at all other CR=8-11. Why you conclude that this is due to the changes of combustion environment caused by longer ignition delay period at too high CR value? What about the rest CR? Please comment.
  4. Lines 258-260. The combustion process in Figure 5e is different from Figures 5a~5d for a compression ratio 12. This trend needs special attention and discussion. The CR=12 can not be named excessive as you increase CR step-by-step. Or may be analysis with CR=13 or 14 can prove or not. Please comment and discuss.
  5. Lines 308-316. Introduction of Summary and Conclusions can be shortened as it almost repeats the previous.
  6. Why there are no conclusion on indicated specific fuel consumption (ISFC), indicated mean effective pressure (IMEP), indicated thermal efficiency (ITE)? Please add to the conclusion on performance parameters.

Author Response

(The authors gave the same response as above.)
